# The Participation of Children with Intellectual Disabilities: Including the Voices of Children and Their Caregivers in India and South Africa

**DOI:** 10.3390/ijerph17186706

**Published:** 2020-09-15

**Authors:** Shakila Dada, Kirsty Bastable, Liezl Schlebusch, Santoshi Halder

**Affiliations:** 1Centre for Augmentative and Alternative Communication, University of Pretoria, Pretoria 0001, South Africa; kgb0071978@gmail.com (K.B.); liezl@treesofhope.co.za (L.S.); 2Department of Education, University of Calcutta, Kolkata 700078, India; santoshi_halder@yahoo.com

**Keywords:** participation, intellectual disabilities, low- and middle-income country, self-report, proxy report, India, South Africa

## Abstract

There is a shortage of research on the participation of children with intellectual disabilities from middle-income countries. Also, most child assessments measure either the child’s or the caregiver’s perceptions of participation. Participation, however, is an amalgamation of both perspectives, as caregivers play a significant role in both accessing and facilitating opportunities for children’s participation. This paper reports on both perceptions—those of children with intellectual disabilities and those of their caregiver, in India and South Africa. A quantitative group comparison was conducted using the Children’s Assessment of Participation and Enjoyment (CAPE) that was translated into Bengali and four South African languages. One hundred child–caregiver dyads from India and 123 pairs from South Africa participated in the study. The results revealed interesting similarities and differences in participation patterns, both between countries and between children and their caregivers. Differences between countries were mostly related to the intensity of participation, with whom, and where participation occurred. Caregiver and child reports differed significantly regarding participation and the enjoyment of activities. This study emphasises the need for consideration of cultural differences when examining participation and suggests that a combined caregiver-and-child-reported approach may provide the broadest perspective on children’s participation.

## 1. Introduction

The right of individuals with disabilities to “full and effective participation and inclusion in society” [1] (p. 6) is enshrined in the United Nations convention on the rights of people with disabilities. As a result, participation is often highlighted as an intervention goal for children who have disabilities. However, the measurement and implementation of this goal have been challenging as participation is bi-directional. That is, participation is both the mechanism for, and the outcome of development [2]. Where children’s development is typical, they participate in activities using behaviours or skills, the complexity of which is fostered and increased within the activities until the skills are mastered [2,3]. For children with disabilities, however, barriers to participation may arise both from individual and environmental factors. As such, the child may be prevented from participating or from participating at the required level for their skills to develop and grow. The barriers experienced by children with disabilities have been described in studies that compared the participation of children with disabilities to that of their peers who were typically developing. These studies identified decreased attendance (or diversity) in active physical, academic, and social activities [4,5,6], and decreased intensity (or frequency) of participation in formal activities outside of school [4,5] for children with disabilities.

In addition to the challenges identified in the research for children with disabilities, several gaps have been identified in the literature on participation. First, the majority of research has been conducted among children with cerebral palsy, while research on other disabilities has been limited [7,8,9,10,11,12,13]. This is despite evidence that intellectual disability has been ranked as one of the most severe and commonly occurring disabilities in children worldwide [14]. Intellectual disability is a pervasive and lifelong condition characterised by significant limitations in both intellectual functioning and adaptive behaviour originating before the age of 18 years [15]. A recent systematic review of the participation of children with intellectual disabilities identified only four studies [16], while a further four studies not included in the review were subsequently also identified [4,17,18,19]. These studies noted more limited participation in active physical and skills-based activities for children with intellectual disabilities [4,16,17,18,19,20] than for typically developing children, and found that participation occurred more frequently in the home setting with adults, rather than in the community with peers [4,19,21].

A second challenge specifically related to children with intellectual disabilities is that the tools used to obtain participation data have thus far been founded on the premise that children’s participation is best understood from their own perspective [4,22]. Hence, assessments of participation rely on only information from the child. However, as described by Nilsson et al. (2015), including the perspectives of adults/children in research is not an either/or scenario but rather a continuum, and the use of only one component on the continuum can limit the depth of the research [23]. For children with intellectual disabilities in particular, the requirement for self-reporting participation can introduce barriers associated with cognitive, linguistic, and communication difficulties [4], which can affect the results reported. Hence, obtaining participation data on a continuum that includes both adult and child input may be beneficial. This position is supported in the ICF-CY where the role of adults in the participation of children who are younger or who have disabilities is described as “integral to understanding participation” [24] (pp. xvi.). This is because for these children, participation opportunities are more likely to be identified by parents or caregivers than by the child themselves [24,25,26].

A third challenge in the literature is that both the development of tools to measure participation and studies on the participation of children have been primarily implemented in high-income countries [27,28,29]. Although the Picture my Participation tool [29] has been developed and validated specifically for use in low-and middle-income countries, as yet no comparative data from this has been published. A lack of research from low- and middle-income countries introduces complexity to the evaluation of participation, as low- and middle-income countries are culturally and economically different from high-income countries [30]. Furthermore, as participation requires the measurement of culturally relevant activities [2], cultural differences may affect the results obtained in respect of participation measures and limit the generalisability of findings.

A fourth concern relating to the dearth of participation research in low- and middle-income countries is evidence that worldwide up to 94.5% of children with epilepsy, intellectual disabilities, vision, or hearing loss live in low and middle-income countries [14,30]. Hence, the data on the participation of children with intellectual disabilities is limited both in extent (research among children with intellectual disabilities) and context (research in low- and middle-income countries).

Although the field of participation research has grown since the introduction of the ICF-CY [24], significant gaps in knowledge remain around children with intellectual disabilities and children from low- and middle-income countries. The current study sought to describe and compare the participation of children with intellectual disabilities from two middle-income countries, India (lower-middle-income) and South Africa (upper-middle-income) [31]. Due to the challenges experienced by children with intellectual disabilities in self-reporting, the data on participation was collected using both caregiver and child reports of participation, which enabled a comparison of the child and caregiver reports to compare the two perspectives for similarities and differences.

The countries for the study were selected because India has one of the highest (≈6%), and South Africa has one of the lowest (≈2.25%) reported prevalence of intellectual disability among middle-income countries [14]. In addition, India and South Africa both have cultures that differ from those of high-income countries. Culture is described as the combination of collective norms, values, experiences and histories of a particular group, which emerges from daily activities in which families and communities are connected [32]. Culture is reported to shape the day-to-day activities that are most important to families and communities [33,34,35]. Both India and South Africa tend towards a collectivist culture in which the individual is an integral component of the community, while high-income countries tend towards a more individualistic culture in which the individual remains conceptually separate from the community [35,36,37,38]. Although India and South Africa are considered collectivist cultures, one of the differences between the two cultures is the presence of the class system that is socially maintained in India. At the same time, South Africa is divided primarily by racial group and economic class [36]. Furthermore, India has a much higher gender gap than South Africa, but South Africa reports higher poverty rates [31,39]. Comparing India and South Africa may provide insight into differences in participation not only between high- and middle-income countries, but also between middle-income countries with different cultures.

## 2. Materials and Methods

### 2.1. Aims

The current research study aimed to (a) describe the participation of children with intellectual disabilities from India and South Africa; (b) compare the participation results between groups (India and South Africa) and respondents (children and caregivers).

### 2.2. Design

The study made use of a multi-factor design, firstly to describe the participation of children with intellectual disabilities, and secondly to compare self- and proxy-reported and country-specific participation data.

### 2.3. Sampling and Participant Selection

Participants for this study were selected using convenience sampling in schools and centres that catered for children who have intellectual disabilities. To participate in the study, children had to be between the ages of six and 21 and scored as having a mild to moderate intellectual disability on the Kaufman Brief Intelligence Test (KBIT) [40]. The children’s home language had to be Bengali, English, Afrikaans, isiZulu, or isiXhosa, as these were the languages into which the Children’s Assessment of Participation and Enjoyment (CAPE) [41] had been translated. If a child’s home language was not the same as the language in which the CAPE [41] was to be administered at their school, then the child needed to have been schooled in the language of the CAPE [41] for at least 1½ years to be included in the study. Caregivers were required to be literate in Bengali, English, Afrikaans, isiZulu, or isiXhosa.

### 2.4. Ethics

India: Ethics approval for the study was obtained from the Ethics Committee of the University of Calcutta, and the appropriate departments and heads of schools or organisations. Participants were recruited from twelve government and non-government associations or schools.

South Africa: Ethics approval was obtained from the Research Committee of the University of Pretoria (GW20160409HS), and permission was obtained from the Department of Education in six provinces in South Africa. Additionally, permission was obtained from school principals or school governing bodies. Participants were recruited from 15 schools (11 government/public schools and four non-government/private schools).

### 2.5. Participants

A total of 100 caregiver-child dyads from India and 123 dyads from South Africa took part in the study. The children ranged in age from five to 18 years (mean = 12.3); 61.3% of them were male, and 38.7% female. Caregivers reporting on their child’s participation were primarily mothers (73.6%), but fathers (15.6%) and other caregivers (10.8%) also responded. Most caregivers had not completed high school and earned less than R4500.00 (approximately €220.00) per month.

Respondents reported their home language as Bengali (37.1%), English (17.2%), Afrikaans (13.6), isiZulu (9.3%), isiXhosa (6.7%), and other languages (16.1%), while the survey was completed in Bengali (43.1%), English (37.1%), Afrikaans (9.5%), isiZulu (5.2%) and isiXhosa (5.2%). The summarised demographic data of the participants is represented in Table 1.

### 2.6. Materials

Data on participation for this study was obtained by administering the Children’s Assessment of Participation and Enjoyment (CAPE) [41]. The CAPE [41] is a self-report questionnaire that has been developed for use with children (with and without disabilities) between the ages of 6–21 years. The CAPE [41] considers 55 activities grouped into eight activity domains: (1) overall; (2) informal activities; (3) formal activities; (4) recreational activities; (5) active physical activities; (6) social activities; (7) skills-based activities; and (8) self-improvement. For each activity domain, five dimensions of participation were measured: (1) diversity; (2) intensity; (3) companionship; (4) location; and (5) enjoyment. The CAPE [41] is typically administered in an interview with the child and takes 45 min to complete [41].

The CAPE [41] was developed and has been shown to have adequate validity and reliability in English [42,43,44]. Internal consistency and test-retest reliability have been confirmed in Dutch [45], Greek [46], Spanish [47,48], Swedish [49], Chinese [50] and Norwegian [51]. However, the need for cultural validation (in addition to translation) has been highlighted by various authors [48,49,50,52]. For this study, permission was obtained by the publishers to translate the CAPE [41] for use in India and South Africa. For India it was translated into Bengali and for South Africa into Afrikaans, isiXhosa and isiZulu. Translation included forward and backward translation as well as the consideration of linguistic, functional and cultural equivalence [53].

For this study, the proxy report version of the CAPE [41] was developed with permission from the publishers, and it was translated following the same procedures as the child’s version [52]. The caregiver version of the CAPE [41] was modified so that the subject of all questions was the caregiver’s child and not the caregiver. The caregiver version was a pen-and-paper version of the CAPE [41].

### 2.7. Data Collection

Caregivers received an information pack from their child’s school. The language of the information pack was based on feedback from the school. Included in the information pack were an information letter about the study, a consent form for caregivers to complete, and the proxy version of the CAPE [41]. Caregivers had to give consent for themselves and their children to be involved in the study. Caregivers independently completed and returned the consent form and proxy version of the CAPE [41] to the school. Children were asked to provide assent to participate in the study on the day of data collection. The researcher administered the CAPE [41] to the children individually at school. The child was asked the questions and their answers were recorded using the methods and visual supports recommended in the manual. The CAPE [41] was administered to the children in the language of the school, and participants were provided with a small token of appreciation for their help (a ruler and an eraser). The researchers were fluent in the language in which they administered the CAPE [41].

### 2.8. Data Analysis

The data for this study was analysed using IBM SPSS version 26 [54]. Demographic data was described statistically using means (reported in percentages). The distribution of the data was assessed and found to be non-normally distributed. Hence, further assessments were non-parametric (Pearson Chi-squared, Fisher’s exact test) to test for group equivalence. Participation data for each participant was summed, while data on intensity, with whom, where and enjoyment was calculated as the mean of all activities participated in. The CAPE [41] data was assessed for internal consistency using Cronbach’s alpha. Between-group statistical correlation of the participation data was conducted using an independent samples Kruskal–Wallis test with a post hoc pairwise comparison using the Bonferroni correction for multiple tests.

## 3. Results

The results of this study are presented below. The internal consistency of items on the CAPE [41] is presented first. This is followed by the self- and caregiver-reported participation of children with intellectual disabilities for India and South Africa. The reported data is then compared across children and caregivers and India and South Africa. Participation is reported overall, in the formal and informal domains, the five activity groups, and the five participation dimensions.

### 3.1. Internal Consistency of the CAPE Data from India and South Africa

The analysis of the data from the CAPE [41] from India and South Africa indicated that the internal consistency of the data was excellent across all domains, for both children and caregivers (0.923 < α > 0.993) [55].

### 3.2. Participation of Children with Intellectual Disabilities

Significant differences were evident in the reported participation of children in India and South Africa, and as reported by children or their caregivers in all areas except participation in self-improvement.

#### 3.2.1. Self-Reported Participation

The self-reported participation of children with intellectual disabilities averaged 27 activities in India and South Africa. Children from India were most likely to participate in activities two to three times a week, with their families, at a relative’s house. In South Africa, however, children were most likely to participate in activities once a week, with other relatives, and at a relative’s house. Children from both countries enjoyed participating in activities “very much”. Participation results across all domains, and dimensions are presented in Table 2.

#### 3.2.2. Caregiver-Reported Participation

Caregivers in India reported that their children with intellectual disabilities participate in 24 activities, compared to 28 activities reported by caregivers in South Africa. Children in India were reported to participate in the activities two to three times a week, while in South Africa, participation once a week was reported. Children in India were reported to participate most often with family, while children in South Africa were reported to participate most often with other relatives. Children in both India and South Africa were most likely to participate in activities at a relative’s house and were reported to enjoy participation “pretty much”. The full participation results are listed in Table 2.

### 3.3. Comparing the Self-and Proxy Reported Participation of Children with Intellectual Disabilities in India and South Africa

The self-reported participation of children with intellectual disabilities in India and South Africa was not significantly different overall, in the informal domain, for social, skills-based, or self-improvement activities. However participation in the formal domain and active physical and recreational activities was significantly different. Intensity, with whom and where participation occurred also differed significantly when self-reported by children in India and South Africa, but enjoyment was not significantly different.

Caregivers in India and South Africa reported differences in participation of their children who have intellectual disabilities overall, in the informal domain and in recreational and skills-based activities. Intensity, with whom and where participation occurred also differed significantly as did enjoyment in a number of areas.

Caregivers and children in India provided similar reports on participation in the informal domain, recreational, active physical, social and self-improvement activities. While caregivers and children in South Africa provided similar reports on participation in all areas except for social activities. No significant differences were evident between most child- and caregiver- reports on intensity, with whom and where participation occurred. However, significant differences were evident in the reports of enjoyment of participation between children and their caregivers in South Africa, but not in India.

## 4. Discussion

This study considered the participation of children with intellectual disabilities in two middle-income countries, India and South Africa. First, the participation in India and South Africa is described (self- and proxy-reported) and considered in relation to results from high income countries. Then the differences between self- and proxy-reports are examined. The discussion concludes with recommendations for practice and future research.

The self-reported participation of children who have intellectual disabilities in India and South Africa shows similarities overall, in the informal domain, social, skills-based and self-improvement activities. These results are also comparable to those from children who have intellectual disabilities in Australia [4], a high-income country. Differences in participation in the formal, recreational, and active physical domains may be associated with culture. These activities fall mostly in the formal domain, and are organised by adults. For example, sports are strongly influenced by the environment and culture. Hence, children in the US are more likely to play baseball and American football, while children in India are more likely to play cricket and hockey, and children in South Africa soccer and rugby. In contrast, socio-emotional development, which is key in informal and social activities has been reported to be similar across different collectivist cultures and individualistic cultures [35], hence the similarities in these areas between India and South Africa are not unexpected.

Differences in the intensity of participation reported between India and South Africa were evident. These differences may be related to where and with whom children participate. In the case of the children from India, a more frequent intensity of participation was reported, but participation also occurred more often at home and with their immediate family. Thus it is plausible that because the child does not have to go anywhere or participate with anyone other than those nearby, participation may occur more frequently. In contrast, the children in South Africa indicated a lower intensity of participation but more often had extended family as participation partners and venues. When compared to results from the study conducted in Australia, participation intensity in South Africa was more similar to that in Australia than that in India. Differences in with whom and where children participate could well relate to cultural or family practices, family support or the availability of resources. It is interesting that in India, where employment rates were significantly higher than in South Africa, participation occurred most often at home, as one might infer that children would attend a daycare or similar while caregivers were at work. This result, however, may suggest a greater level of in home support for caregivers in India, than for caregivers in South Africa.

Enjoyment of participation was similar when self-reported between India and South Africa, except for self-improvement activities. The mean scores for enjoyment indicate that both groups of children enjoyed these activities “very much”. These results concur with other studies on participation, which propose that children, regardless of disability, enjoy participating in activities to a high degree [52].

The proxy-reported participation and enjoyment of children differed significantly between India and South Africa in all areas, except for formal, active physical and social activities. The similarities in participation in formal, active-physical, and social activities between caregivers is of interest, as it is in contrast to the differences reported by their children. A possible explanation of these differences may be related to the number of activities that children participated in. Overall caregivers in India reported fewer activities that their children participated in than caregivers in South Africa. If the activities which were not reported as participated in fell into the informal domain or were clustered in recreational, and skills-based activities a decrease in participation in these areas and the measurement of enjoyment in these areas may be seen. An alternative explanation relates to the historic educational contexts of India and South Africa. Both countries were historically British colonies, whose education systems were obligated to follow a British model, features of which are maintained today. As caregivers would have been educated in this model, it is possible that their perceptions of formal, active physical and social activities have been informed by similar constructs within the education systems, while the informal domain has been formed to a greater extent by the local culture.

Caregivers’ reports on intensity of participation, with whom and where participation occurred, mirrored those of their children, showing significant differences in each of these areas. The reporting of Enjoyment of participation showed a similar pattern of significant differences to participation.

When the self- and proxy-reported participation data is compared, differences were evident in India for participation overall, in the formal domain, active physical and skill-based activities. In South Africa however only social activities were reported differently. Reasons for the different directions of reporting are not known but could relate to the child’s preferences or independence. Caregivers may only perceive participation to be occurring when their child has chosen to participate, or is able to participate independently. In contrast, children may report on “being there”, regardless of their choice or role in the activity [56]. In addition, different cultural norms may impact the activities available to children. For example, within Indian culture, a person’s position is strongly determined by their caste, and the activities families may access are determined by this social structure [36,37,38]. In South Africa, on the other hand, the provision of activities is more likely to be linked to an educational or economic opportunity. Other differences not explored in this paper could also be the availability of support from others, the number of children in the household, or the cultural beliefs of the caregivers regarding children with disabilities, their needs, rights and abilities [57,58].

This paper expected to report significant differences in the reporting of enjoyment for self and proxy-reports. This was based on studies that have suggested that proxy reports vary from direct-reports in areas of non-observable functioning, for example emotions [59,60,61,62]. Hence, the lack of difference in the reporting of enjoyment of participation between children and caregivers in India requires consideration. It is possible that differences in the expression of enjoyment could be linked to communication styles from India and South Africa. As reported in relation to socioemotional skills, adolescents from South Africa were reported to be more likely to have medium communication skills than counterparts from Malaysia, who were more likely to have high communication skills (an Asian collectivist culture). If this pattern is repeated between India and South Africa, it would suggest that children in India communicate with their caregivers at a different level than do children in South Africa, which could impact the likelihood of emotions, which are not clearly visible being recognised [35] and accurately reported.

Overall, the significant differences in participation from children in India and South Africa highlight the need for increased research from low- and middle-income countries and for an understanding that participation is culturally biased. Hence there is a need for participation assessment and intervention to take cultural norms, values, and differences into account. In addition, there are significant differences in self- and proxy-reports on participation, but the differences do not negate the consideration of either perspective. Instead, if perspective, as described by Nilsson et al. [23], is considered, the combination of child and proxy reports could provide more comprehensive participation information, particularly where children experience difficulty in communicating.

### 4.1. Limitations

This study is limited by a lack of comparison group of children who have typical development in either country. Without this, it is not possible to ascertain if differences in participation are culturally based or related to the children’s disabilities. In addition, it is not possible to determine if the children in India and South Africa who have intellectual disabilities are able to have “full and effective participation and inclusion in society” [1] (p. 6). A further limitation of this study is the lack of comparative data from high-income countries. Although results from Australia are referred to, without the actual data for full analysis, the comparison between the participation of children with intellectual disabilities from middle- and high-income countries remains superficial.

### 4.2. Recommendations for Practice and Future Research

For interventionists in the field, particularly those working in diverse societies, the participation of children is a key aim. However, the measurement of participation needs careful consideration due to the sensitivity of participation to cultural differences, and the need to include both self- and proxy-reported participation data. Goal setting for intervention needs to be driven by the child’s, family such that their own culture is expressed in their child’s participation.

Further research on the reasons for the differences between the participation of children in South Africa and India is required, in particular concerning environmental factors such as culture, social support and available resources. Also, further research on participation of children who have typical development in low-income and other middle-income countries would add to the understanding of participation in different cultures. Further research on the validity of the CAPE [41] in low- and middle-income settings is also required.

## 5. Conclusions

Overall, the patterns of participation of children with intellectual disabilities in India and South Africa showed a number of similarities. However, differences in patterns of participation could be related to cultural differences between the two countries. Caregiver and child reports on participation also showed differences that could be culturally based. The results of this study support the use of proxy reporting for measuring participation as a valid strategy when used in combination with child reporting, specifically of perspectives and emotions. Further research on the participation of children in low- and middle-income countries is recommended.

## Figures and Tables

**Table 1 ijerph-17-06706-t001:** Demographic data of participants from India and South Africa.

	India	South Africa	Combined Data	Equivalence ^1^ *p*-Value
Caregiver-child dyads	*n* = 100	*n* = 132	232	
Child age (years)	mean = 11.9 (SD: 2.5)	mean = 12.7 (SD: 2.6)	mean = 12.3	0.000 ^2^
Sex				
Male	66.0	57.8	61.3	0.44 ^4^
Female	34.0	42.2	38.7	
Additional impairments (%) ^3^	12.6	13.0	25.6	0.17 ^4^
Home language				
	Bengali: 37.1 Hindi: 5.2 Other: 2.0	English: 17.2 Afrikaans: 13.6 isiZulu: 9.3 isixhosa: 6.7 SiSwati: 1.3 Sesotho: 2.6 Sepedi: 2.6 Other: 2.3 Setswana: 1.7		
Survey language (%) ^3^				
Bengali	43.1			
English		37.1		
Afrikaans		9.5		
isiZulu		5.2		
isiXhosa		5.2		
Caregiver respondent (%) ^3^				
Mother	33.8	39.8	73.6	0.129 ^4^
Father	6.1	9.5	15.6	
Other	3.5	7.4	10.8	
Caregiver education (%) ^3^				
Grade 11 or less	23.1	17.8	40.9	0.000 ^2^
Grade 12	7.1	17.8	24.9	
Degree	12.0	8.9	20.9	
Other	2.2	11.1	13.3	
Household income (%) ^3^				
<R4500 (≈€220)/month	1.8	26.2	28.0	0.000 ^2^
R4501-R12500 (≈€600)/month	4.4	12.4	16.9	
R12501-R30000 (≈€1500)/month	11.6	7.6	19.1	
R30001-R52000 (≈€2500)/month	5.8	3.6	9.3	
R52001-R70000 (≈€3370)/month	6.2	3.1	9.3	
>R70001 (≈€3370)/month	14.7	2.7	17.3	

Notes: ^1^ Group equivalence between India and South Africa; ^2^ Pearson chi-square *p* < 0.05; ^3^ Percentages may not add up to 100 due to rounding; ^4^ Fisher’s exact test—one-sided.

**Table 2 ijerph-17-06706-t002:** Self-reported and caregiver-reported participation of children with intellectual disabilities in India and South Africa (Mean).

	India		South Africa		All Participants		Significance ^†^
	Child	Caregiver	Child	Caregiver	Child	Caregiver	*p* < 0.05
Participation domains and activities ^1^							
Overall ^i^	27.02	23.98 ^a^	26.77	28.36	26.88	26.47 ^b^	0.001
Informal ^ii^	20.70	19.57	22.01	23.04	21.44	21.54 ^b^	0.000
Formal ^iii^	6.32	4.50 ^a^	4.91	5.44	5.53 ^b^	5.04	0.000
Recreational ^iv^	6.86	6.25	8.21	8.33	7.63 ^b^	7.43 ^b^	0.000
Active physical ^v^	5.45	4.01 ^a^	4.36	4.95	4.84 ^b^	4.54	0.000
Social ^vi^	6.52	6.94	6.23	6.97 ^a^	6.35	6.96	0.008
Skills-based ^vi^	3.70	2.43 ^a^	3.33	3.53	3.49	3.06 ^b^	0.000
Self-improvement ^vi^	4.71	4.61	5.11	5.06	4.94	4.87	0.084
Intensity of participation ^2^							
Overall ^i^	5.92	5.94	4.98	4.98	5.38 ^b^	5.40 ^b^	0.000
Informal ^ii^	5.93	5.89	5.01	5.02	5.41 ^b^	5.39 ^b^	0.000
Formal ^iii^	5.92	6.30	4.97	4.83	5.39 ^b^	5.46 ^b^	0.000
Recreational ^iv^	6.33	6.33	5.48	5.34	5.85 ^b^	5.77 ^b^	0.000
Active physical ^v^	5.82	5.81	4.64	4.86	5.16 ^b^	5.28 ^b^	0.000
Social ^vi^	5.33	5.09	4.38	4.52	4.79 ^b^	4.76 ^b^	0.000
Skills-based ^vi^	6.56	7.29 ^a^	5.12	4.82	5.75 ^b^	5.88 ^b^	0.000
Self-improvement ^vi^	6.14	6.32	5.15	5.27	5.58 ^b^	5.73 ^b^	0.000
With whom participation occurred ^3^							
Overall ^i^	1.68	1.57	2.40	2.62	2.09 ^b^	2.16 ^b^	0.000
Informal ^ii^	1.61	1.51	2.29	2.46	2.00 ^b^	2.05 ^b^	0.000
Formal ^iii^	1.89	1.77	2.90	3.36	2.46 ^b^	2.68 ^b^	0.000
Recreational ^iv^	1.53	1.34	2.26	2.30	1.94 ^b^	1.89 ^b^	0.000
Active physical ^v^	1.67	1.64 ^a^	2.83	3.44	2.32 ^b^	2.65 ^b^	0.000
Social ^vi^	1.72	1.65	2.47	2.52	2.15 ^b^	2.14 ^b^	0.000
Skills-based ^vi^	1.88	1.65	2.86	3.34	2.43 ^b^	2.61 ^b^	0.000
Self-improvement ^vi^	1.63	1.58	1.98	2.35	1.83 ^b^	2.02 ^b^	0.000
Where participation occurred ^4^							
Overall ^i^	1.79	1.83	2.53	2.43	2.21 ^b^	2.18 ^b^	0.000
Informal ^ii^	1.66	1.71	2.36	2.23	2.06 ^b^	2.01 ^b^	0.000
Formal ^iii^	2.23	2.36	3.49	3.40	2.94 ^b^	2.95 ^b^	0.000
Recreational ^iv^	1.44	1.41	1.87	1.83	1.68 ^b^	1.65 ^b^	0.000
Active physical ^v^	1.89	2.09	2.95	2.97 ^a^	2.49 ^b^	2.58 ^b^	0.000
Social ^vi^	2.00	2.11	2.70	2.51	2.40 ^b^	2.34 ^b^	0.000
Skills-based ^vi^	2.25	2.11	2.94	3.11	2.64 ^b^	2.67 ^b^	0.000
Self-improvement ^vi^	1.55	1.59	2.74	2.64	2.22 ^b^	2.18 ^b^	0.000
Enjoyment of participation ^5^							
Overall ^i^	4.34	4.23	4.29	3.92 ^a^	4.31	4.05 ^b^	0.000
Informal ^ii^	4.35	4.23	4.28	3.89 ^a^	4.31	4.04 ^b^	0.000
Formal ^iii^	4.32	4.22	4.31	4.04 ^a^	4.32	4.12	0.000
Recreational ^iv^	4.35	4.20	4.35	3.87 ^a^	4.35	4.02 ^b^	0.000
Active physical ^v^	4.35	4.20	4.35	3.97 ^a^	4.35	4.07	0.000
Social ^vi^	4.36	4.25	4.43	4.15 ^a^	4.40	4.19	0.000
Skills-based ^vi^	4.23	4.25 ^a^	4.25	3.85	4.25	3.97 ^b^	0.000
Self-improvement ^vi^	4.32	4.12	3.92	3.39 ^a^	4.09 ^b^	3.70 ^b^	0.000

Notes: ^†^ Independent Samples Kruskal–Wallis test (*p* < 0.05). Post hoc pairwise comparison with Bonferroni correction for multiple tests (*p* < 0.05): ^a^ Significant caregiver-child difference; ^b^ Significant India-South Africa difference; ^1^ Participation = mean number of activities attended out of ^i^ 55 activities, ^ii^ 40 activities, ^iii^ 15 activities, ^iv^ 12 activities, ^v^ 13 activities, ^vi^ 10 activities; ^2^ Intensity: 0 = never, 1 = once in 4 months, 2 = twice in 4 months, 3 = once a month, 4 = 2–3 times a month, 5 = once a week, 6 = 2–3 times a week, 7 = once a day; ^3^ With whom: 1 = alone, 2 = with family (parents, siblings), 3 = with other relatives (grandparents, uncles, cousins, etc.), 4 = with friends, 5 = with others; ^4^ Where: 1 = at home, 2 = at a relative’s house, 3 = in your neighbourhood, 4 = at school, 5 = in your community,.6 = beyond your community; ^5^ Enjoyment: 1 = not at all, 2 = sort of/somewhat, 3 = pretty much, 4 = very much, 5 = love it!

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
