# Peer review of "The Participation of Children with Intellectual Disabilities: Including the Voices of Children and Their Caregivers in India and South Africa"

_ijerph, 2020, doi:10.3390/ijerph17186706_

Round 1

Reviewer 1 Report

General comments:

This study was to investigate the perceptions of participation of children with intellectual disabilities and their parents from middle-income countries. The participants are One hundred children-caregiver dyads from India and 123 pairs from South Africa with The Children’s Assessment of Participation and Enjoyment (CAPE), which was translated into Bengali and four South African languages. The results revealed interesting similarities and differences in participation patterns, both between children and their caregivers as pairs, and between countries. The caregiver and child reports differed significantly regarding the enjoyment of activities. Significant differences between countries were related to the intensity of participation, with whom, and where participation occurred. The study procedures are innovative for comparing both perspectives and providing larger information about both types of opinions.

The major concern will be the analysis strategy. Multiple comparisons using statistics in the same sample would face the problem of inflation of p-value, betters using factors design, and there to avoid multiple comparisons. https://ekja.org/journal/view.php?number=8476

Consider this article about post-hoc analysis about multiple comparisons

.

Specific comments:

Abstract:. .                                                 

  1. P,1, In the abstract, please mention the overall research method or strategy, qualitative or quantitative?
  2. P1,line28, May not all developmental, some participation is not related to age. I suggest changing specifically to the “ intervention” goal.
  3. P2,line58, Why need proxy report and children’s report, in this paper you mean proxy’s report is a supplementary or support or substitute.
  4. P2,line67, There are also some in low-middle income like the “Picture My Participation for measuring participation” (Arvidsson, P., Dada, S., Granlund, M., Imms, C., Bornman, J., Elliott, C., & Huus, K. (2020). Content validity and usefulness of Picture My Participation for measuring participation in children with and without intellectual disability in South Africa and Sweden. Scandinavian Journal of Occupational Therapy, 27(5), 336-348. doi:10.1080/11038128.2019.1645878)

  1. P2,line84, did you mean a comparison between …….is your purpose? If so, it should be matched in the abstract and methods.
  2. P3, Line 102, Purpose can be more collective and comprehensive. Such as “1) to measure the participation …. ‘(not the purposes and you must measure anyway, it’s more like a statistics in the methods, I think it can be omitted in purposes). “2)compare the participation between groups (two countries) and respondents (parents and children) (there may be the main effect (groups and respondents) or interaction (moderator effect).” The impact effects of countries on participation would is different from respondents and vice versa.
  3. P3, line 120, A two factors design? If parents and children were all pairs, we should consider some other statistics for pairs.
  4. P5, line 180, Does the interviewers (researchers) know all the languages, or they have a group to provide specific interview?
  5. P5, line 190, With interview guide not self-complete the questions, how to deal with the comprehension or expression difficulties, such as using proxy as help, if so the answers of parents and children may not total independent, that’s why we should compare pair statistics. Or some assistant instructions, such as a flash card, or there are missing data form the answers of the child?
  6. P6, Line 208, may combine the table that the uniqueness in this paper of the parent-child dyad can be prominent. Illustrated as below:

Indian

South Africa

Different between all parents and child

Interaction and post hoc

Parent

child

Parent

child

Participation domains

.

.

.

  1. P 9, table 4, if these values in the table are p-value, it should be noted. The * sign seems redundant, and it means the same thing, better integrated into the previous table.
  2. Discussion: Well discussed. But if the analysis change to a more focused on purpose, it may have some interaction effect when you compare the results between counties and respondents.

Reviewer 2 Report

In my opinion, this is an interesting contribution, which highlights a very important aspect of the quality of life of children with disabilities (participation) in two significant middle-income countries.

The research is appropriately designed and presented, I have only some small suggestions, which I think could improve the paper:

  1. Firstly, I suggest including in the introduction some reference to the UN Convention on the Rights of Persons with Disabilities. This is important not only because the Convention is today the most important international framework on disability policies, which should always be taken into account by researchers, but also because, on the specific issue dealt with in this paper, the Convention provides a very valuable support to stress and highlight the importance of participation. This is already shown by the definition of disability provided in the Convention, which makes reference to the barriers to social participation (Preamble and Article 1); moreover, “full and effective participation and inclusion in society” is one of the principles of the Convention (article 3), and several articles are devoted to participation in various fields (for example, articles 19, 29, 30). The Convention also pays attention to children with disabilities: article 7.
  2. Although I recognize that this goes beyond the specific purpose of this paper, I think that the discussion should include some assessment of the degree of participation of children with disabilities in the countries studied. This paper tells us how much and where the children with intellectual disabilities of India and South Africa participate, but it does not tell us if that is enough or not, good or bad. Thus, the information provided is useful, but certainly limited. Maybe the comparison with Australia could help to provide that assessment. And if this assessment were made, maybe the article could also include some recommendations for disability policies. In fact, in line 276 the authors announce that “the discussion concludes with recommendations for practice and future research”, but afterwards one only finds recommendations for future research, but not for practice.
  3. Finally, although English language and spelling of the article seems generally fine, I think a revision could be useful to improve the style, for example avoiding the repetition of words in the same paragraph.
  4. A very specific suggestion concerning language: I think the adjective disabled, used in lines 62 and 63, should be avoided, and replaced by the form “children with disabilities” or “children which have a disability”.

Round 2

Reviewer 1 Report

The article is now more understandable and clearly on the linkage between purposes and statistics.

But the two table are still disconcerting.

The denotes of the two tables on the no.of items of activity types and scores may move to the material into text. The significance of pair comparison can be noted as superscripts. Table 3 could make the readers more confused because they should mark the significances. I described in attached pdf with the words and letters in red.
